



# Physiological control on carbon isotope fractionation in marine phytoplankton

Karen M. Brandenburg[1], Björn Rost[2,3], Dedmer B. Van de Waal[4], Mirja Hoins[1,2], Appy Sluijs[1]

[1]Department of Earth Sciences, Faculty of Geosciences, Utrecht University, Princetonlaan 8a, 3584 CB Utrecht, the Netherlands

[2]Department of Marine Biogeoscience, Alfred Wegener Institute (AWI), Helmholtz Centre for Polar and Marine Research, Am Handelshafen 12, 27570 Bremerhaven, Germany

[3]Faculty of Biology/Chemistry, University of Bremen, Leobener Strasse, 28359 Bremen, Germany

[4]Department of Aquatic Ecology, Netherlands Institute of Ecology (NIOO-KNAW), Droevendaalsesteeg 10, 6708 PB Wageningen, the Netherlands

*Correspondence to*: Karen M. Brandenburg (k.m.brandenburg@uu.nl)

**Abstract.** One of the great challenges in biogeochemical research over the past half a century has been to quantify and understand the mechanisms underlying stable carbon isotope fractionation ($\varepsilon_p$) in phytoplankton in response to changing $p$CO$_2$. Partly, this interest is grounded in the use of fossil photosynthetic organism remains as a proxy for past atmospheric CO$_2$ concentrations. Phytoplankton organic carbon is depleted in $^{13}$C compared to its source because of kinetic fractionation by the enzyme RubisCO during photosynthetic carbon fixation, as well as through physiological pathways upstream of RubisCO. Moreover, other factors such as nutrient limitation, variations in light regime as well as phytoplankton culturing systems and inorganic carbon manipulation approaches may confound the influence of CO$_2$ on $\varepsilon_p$. Here, based on experimental data compiled from the literature, we assess which underlying physiological processes cause the observed differences in $\varepsilon_p$ for various phytoplankton groups in response to C-demand/C-supply and test potential confounding factors. Culturing approaches and methods of carbonate chemistry manipulation were found to best explain the differences in $\varepsilon_p$ between studies, although daylength was an important predictor for $\varepsilon_p$ in haptophytes. Extrapolating results from culturing experiments to natural environments and for proxy applications therefore requires caution, and it should be carefully considered whether culture methods and experimental conditions are representative of natural environments.

## 1 Introduction

Understanding of past climates, in particular variations in atmospheric CO$_2$ concentrations and concomitant temperatures, may help to improve climate models and constrain the global temperature response to projected CO$_2$ rise (Rohling et al., 2012; Zhu et al., 2020; Tierney et al., 2020). Reconstructions of past CO$_2$ concentrations beyond the reach of ice cores rely on proxy estimates. These are based on biogeochemical relations between CO$_2$ concentrations in the atmosphere and the chemical or morphological properties of biogenic carbonates, other minerals, fossil leaves or various types of organic matter that can be found in sediments (Foster et al., 2017; Macdonald, 2020). All these proxies are based on assumptions and exhibit large uncertainties that are ideally constrained iteratively.



One line of proxies uses the $CO_2$-dependence of $^{13}C$ fractionation during photosynthetic carbon fixation in phytoplankton
(O'Leary, 1984; Sharkey and Berry, 1985; Farquhar et al., 1989). Several components found in sediments have been proposed
for this, including bulk organic matter (Hayes et al., 1999) and algae-derived molecules, such as porphyrins (Freeman and
Hayes, 1992) and phytane (Bice et al., 2006) produced by all photosynthetic organisms. Also, the potential of more specific
proxies has been tested, including alkenones originating from coccolithophores (Jasper and Hayes, 1990) and resting cysts
produced by dinoflagellates (Hoins et al., 2015; Sluijs et al., 2018). The $\delta^{13}C$ signal of various algal remains is thought to
follow the $\delta^{13}C$ signal of dissolved inorganic carbon (DIC), in particular the $\delta^{13}C$ signal of $CO_2$, modulated by $CO_2$-dependent
fractionation during photosynthetic carbon fixation. Therefore, ultimately, the $\delta^{13}C$ signal in algal fossil remains may be used
for estimating atmospheric $CO_2$ levels through geological time. Accurate use of this proxy relies on the mechanistic
understanding of carbon isotope fractionation ($\varepsilon_p$) in phytoplankton, which is obtained by different culturing approaches and
assays targeting relevant physiological pathways.

Fractionation occurs during fixation of $CO_2$ by ribulose-1,5-bisphosphate carboxylase/ oxygenase (RubisCO) (Raven and
Johnston, 1991), and is further dependent on the C-supply to this enzyme as well as the C-demand of the cells (Rau et al.,
1996; Bidigare et al., 1997; Hoins et al., 2016). RubisCO discriminates against the $^{13}C$ isotope resulting in biomass being $^{13}C$-
depleted relative to its source $CO_2$. In higher plants, the intrinsic fractionation value of RubisCO ($\varepsilon_f$) is estimated to be between
26-30‰ (Roeske and O'Leary, 1984; McNevin et al., 2007). However, $\varepsilon_f$ can differ between phytoplankton taxa and species
(e.g., Maberly et al., 1992; McNevin et al., 2007), and is indeed an important source of variation in $^{13}C$ fractionation among
phytoplankton groups. Several catalytically and phylogenetically distinct forms of RubisCO in phytoplankton exist, including
Forms IA, IB, ID and II (Whitney et al., 2011; Tabita et al., 2008). Direct *in vitro* measurements of $\varepsilon_f$ yielded values of ~11‰
for the haptophyte *Emiliania huxleyi* (Boller et al., 2011) and ~18.5‰ for the diatom *Skeletonema costatum* (Boller et al.,
2015). Much higher fractionation values have been estimated from *in vivo* experiments under nitrate-limited conditions, with
values as high as ~25‰ for the diatoms *Phaeodactylum tricornutum* and *Porosira glacialis,* and for *E. huxleyi* (Popp et al.,
1998), and ~27‰ for the dinoflagellate *Alexandrium tamarense* (Wilkes et al., 2017).

These large differences between phytoplankton groups and across treatments point towards physiological processes that can
affect fractionation, notably those involved in so-called carbon concentrating mechanisms (CCMs). CCMs have evolved over
time as a response to declining atmospheric $CO_2$ concentrations to ensure effective carboxylation in the vicinity of RubisCO
in oxygenated waters (Giordano et al., 2005). Phytoplankton CCMs comprise a variety of physiological adaptations, and
include active uptake of $CO_2$ and $HCO_3^-$, the use of carbonic anhydrase (CA) to accelerate the interconversion between $CO_2$
and $HCO_3^-$, and ways to minimize the $CO_2$ efflux from the cell (Badger et al., 1998; Reinfelder, 2011; Rokitta et al., 2022).
These processes can strongly influence $^{13}C$ fractionation patterns of phytoplankton (Sharkey and Berry, 1985). For instance,
$HCO_3^-$ is enriched in $^{13}C$ relative to $CO_2$ (by ~10‰), and a high uptake and assimilation of $HCO_3^-$ can therefore lower apparent
$^{13}C$ fractionation values. In addition, alterations in the $CO_2$ efflux over total carbon uptake (i.e., leakage) also affect $^{13}C$
fractionation, as faster replenishment of the intracellular $CO_2$ pool prevents a build-up of $^{13}CO_2$ and thus allows RubisCO to
fully express its intrinsic fractionation. Different modes of CCMs are employed by different phytoplankton species, likely



attributing to species-specific or group-specific differences in $^{13}C$ fractionation (Badger et al., 1998; Van de Waal et al., 2019; Tortell, 2000).

The observed differences in $\varepsilon_p$ between nutrient-limited and nutrient-replete cultures have been attributed to differences in the regulation of carbon uptake relative to carbon fixation (Laws et al., 2001). This variation may, at least partly, be caused by culturing methods, as chemostat cultures that were limited by nutrients or light showed similar responses of $\varepsilon_p$ to changes in $CO_2$, while responses in light-controlled dilute batch cultures were markedly different (Laws et al., 2001). Likewise, discrepancies in measured $\delta^{13}C$ values in different species of coccolithophores have been ascribed to varying culture methods,

in particular to methods of $CO_2$ manipulation (Liu et al., 2018; Hermoso et al., 2016). A recent study furthermore attributed differences in apparent fractionation to a regulatory CCM pathway upstream of RubisCO (Wilkes and Pearson, 2019). This pathway was suggested to alleviate excess photon flux when cells are nutrient limited by shunting energy towards carbon uptake and hydroxylation reactions that increase $\varepsilon_p$.

Here, we aim to elucidate which underlying physiological processes cause the observed differences in $^{13}C$ fractionation in

phytoplankton under different $CO_2$ concentrations, and how this is influenced by experimental settings. To this end, we collected data from all available culture studies, including a range of phytoplankton species from different groups, and evaluated systematic trends and offsets in $^{13}C$ fractionation as a function of environmental, physiological, and experimental factors. This analysis compares the drivers behind phytoplankton $^{13}C$ fractionation, assesses relations between $^{13}C$ fractionation and culturing settings, and discusses implications for proxy development.

## 2 Material & Methods

### 2.1 Literature review

We compiled data on $^{13}C$ fractionation ($\varepsilon_p$) in phytoplankton species under a range of $CO_2$ concentrations and experimental conditions. A literature search was performed in Web of Science (https://www.webofknowledge.com/) using the query ("phytoplankton" OR "algae" OR "microalgae" OR "picoplankton") AND ("climate change" OR "ocean acidification" OR

"CO2" OR "carbon dioxide" OR "global change" OR "pCO2" OR "carbonate chemistry") AND ("13C fractionation" OR "εp" OR "carbon isotope" OR "isotope fractionation") on 25-02-2020. Data on $^{13}C$ fractionation, growth rates ($\mu$), and particulate organic carbon (POC) content under different experimental conditions and $CO_2$ concentrations were extracted using Engauge software when needed (Mitchell et al., 1991). To get an estimate of the carbon demands of the cells, we calculated POC production by multiplying the POC content with the instantaneous growth rate ($\mu_i$). Using $\mu_i$, we yield POC production that

corresponds to the carbon fixation during the photoperiod (Riebesell et al., 2000b, a; Rost et al., 2002; Burkhardt et al., 1999a), and therefore corrects for difference in daylength between studies. In addition, information was extracted on experimental settings (i.e., irradiance, light-dark cycle, salinity, temperature, nutrients), culturing approach (i.e., batch, chemostat, dilute batch, dilute chemostat), and type of carbonate chemistry manipulation resulting in different concentrations of dissolved inorganic carbon (DIC; i.e., aeration of culture with $CO_2$, pre-aeration of culture medium with $CO_2$) or total alkalinity (TA;





i.e., acid/base addition). Under non-limiting growth conditions, $\delta^{13}C$ of phytoplankton cells was measured during the exponential growth phase. The database includes only marine and estuarine phytoplankton species, with data acquired through single species culture experiments.

## 2.2 Statistical analyses

All analyses were performed in R version 4.0.3 (R Core Team, 2020). Significant differences in $\varepsilon_p$ between different
experimental conditions and culturing methods were calculated by means of a linear model followed by pairwise comparisons (Tukey method). To assess the relationship between POC production/$CO_2$ and $\varepsilon_p$, a linear model was fitted to the data for each of the distinct phytoplankton groups, and for each of the distinct species and study combinations. Data on POC production/$CO_2$ was first log transformed, as this improved normality. To assess which of the influential conditions (i.e. nutrient conditions, carbonate chemistry manipulation method, culture approach, irradiance or light-dark cycle) could best explain the variation in
$\varepsilon_p$, along with POC production/$CO_2$, we compared different models using the lmer function in R from the package "lme4" (Bates et al., 2015). In these models, POC production/$CO_2$ and one of the influential conditions were fitted as fixed effects, including interaction terms, while species was fitted as a random effect for each of the distinct phytoplankton groups (excluding cyanobacteria due to lack of data). Models were subsequently compared based on their Akaike Information Criterion (AIC) and Bayesian Information Criterion (BIC).

# 3 Results

## 3.1 Dataset on $^{13}C$ fractionation

Our literature search yielded a total of 509 results, first titles and subsequently abstracts were reviewed, which led to a selection of 77 publications for screening. After careful screening for suitability, a total of 25 publications, containing 58 unique datasets, were included in our database. It contains data on four of the major marine phytoplankton groups, namely dinoflagellates (15
datasets), diatoms (24 datasets), haptophytes (17 datasets) and cyanobacteria (2 datasets).

Across all phytoplankton groups, there is a negative log-linear relationship between $^{13}C$ fractionation ($\varepsilon_p$) and POC production over $CO_2$ supply (Fig. 1). This relationship is also apparent in each phytoplankton group (Fig. 2), although the slope of this curve varies between groups, and also strongly between species and studies (see also Fig. 3). As not all studies reported POC contents per cell, we also tested $\mu_i/CO_2$ supply to assess more species (especially for diatoms), finding similar pattern as POC
production/$CO_2$ supply (Fig. S1).

## 3.1 Experimental settings and $^{13}C$ fractionation

Some of the variation in $\varepsilon_p$ can be explained by the different experimental settings between the studies (Fig. 4). For instance, phytoplankton grown under nitrogen limitation and lower temperatures show higher $\varepsilon_p$ than those grown under light-controlled or non-limiting growth conditions (Fig. 4d). However, $^{13}C$ fractionation also varied across the different types of carbonate



chemistry manipulations and culturing approaches (Fig. 4a, c). Closed systems (i.e., pre-aeration with $CO_2$ and acid/base addition) had lower $\varepsilon_p$ values than open systems (continuous aeration with $CO_2$), and cultures that were grown in chemostats with high biomass had higher overall $\varepsilon_p$ values than those grown in dilute cultures. Moreover, light-dark cycle also strongly influences [13]C fractionation, with cultures that experience continuous irradiation having higher $\varepsilon_p$ values than cultures that are exposed to a dark-cycle (Fig. 4b). This was especially apparent for haptophytes and dinoflagellates, where cultures with

continuous light grown in nitrogen-limited chemostats had higher $\varepsilon_p$ values than those with a dark-cycle grown under replete dilute batch conditions (Fig. 2).

Some confounding experimental conditions across studies appear in our database. Notably, nutrient limitation experiments are almost always performed in chemostats with continuous aeration and without a light-dark cycle. In addition, non-limiting or light-controlled culture studies, with a light-dark cycle, are almost entirely performed using a dilute batch, with pre-aeration

or acid/base addition (Fig. 4). To tease apart which of these confounding factors (i.e., nutrient conditions, type of carbonate chemistry manipulation, culturing approach, and light-dark cycle) can best explain the differences in $\varepsilon_p$, besides C-demand/C-supply, we compared different models including POC production/$CO_2$ concentration and one of the variables for each of the distinct phytoplankton groups (Table S2-4). For haptophytes, inclusion of light-dark cycle could best explain the data (AIC 711 and BIC 738), while culturing approach yielded the best results for dinoflagellates (AIC 490 and BIC 520), and for diatoms

this was either culturing approach (AIC 600) or method of carbonate chemistry manipulation (BIC 623).

## 4 Discussion

In our analyses of the current literature on $\varepsilon_p$ responses, we observed a high dependence on C-demand/C-supply across and within different phytoplankton groups (Fig. 1, 2). This correction step for C-demand is essential, as already identified in previous work, because different growth rates and cellular C contents reflect the different C requirements of phytoplankton

cells (Rau et al., 1996; Bidigare et al., 1997; Hoins et al., 2016). Variation in the $\varepsilon_p$ relationship with POC production/$CO_2$ was, however, observed between the different species and studies (Fig. 3). Next to species-specific differences, this may be attributed to the contrasting experimental settings and culture methods. In the following, we will discuss the variation in fractionation patterns between the phytoplankton species and groups, highlighting the potential role of CCMs, how different experimental settings may result in isotopic disequilibria conditions, and the implications for $CO_2$ proxies based on carbon

isotope fractionation.

### 4.1 Fractionation patterns and underlying processes

Across and within phytoplankton groups, the relationship between $\varepsilon_p$ and POC production/$CO_2$ follows a decay function (i.e. see untransformed data Fig. S2), which highlights the active role of CCMs in C uptake in all groups. If species relied on diffusive $CO_2$ uptake alone, a more linear relationship can be expected. The presence of CCMs is further supported by some

of the low $\varepsilon_p$ signals (Fig. 1), indicating a higher contribution of $HCO_3^-$ to C fixation and/or decreased leakage. Intrinsic RubisCO fractionation values ($\varepsilon_f$) of 18‰ and 11‰ were measured in diatoms and haptophytes, respectively (Boller et al.,



2011, 2015). $\varepsilon_p$ values exceeding $\varepsilon_f$ for both of these groups, and possibly also for dinoflagellates, remain therefore puzzling and indicate fractionation steps occurring upstream of RubisCO (Wilkes and Pearson, 2019).

In cyanobacteria, $CO_2$ fixation by RubisCO takes place in the carboxysome, which is a distinct cellular compartment. The
membrane of this compartment prevents diffusion of $CO_2$, while it is permeable for $HCO_3^-$ which is converted to $CO_2$ via carboxysomal CA, thereby accumulating $CO_2$ in the vicinity of RubisCO (Espie and Kimber, 2011; Dou et al., 2008; Price et al., 2008). To prevent $CO_2$ efflux out of the cell, and likewise facilitate diffusive $CO_2$ uptake, cytosolic $CO_2$ is actively converted to $HCO_3^-$ by the NAD(P)H dehydrogenase (NDH) complex in the cytoplasm (Price et al., 2002; Maeda et al., 2002). It was proposed that these specific processes modify and in fact raise $\varepsilon_p$ values in cyanobacteria: A strong disequilibrium in
the cytosol may, for instance, favor a unidirectional conversion of $CO_2$ to $HCO_3^-$ that would result in an additional fractionation step of at least ~13‰ (O'Leary et al., 1992), and potentially up to 20-33‰ (Zeebe and Wolf-Gladrow, 2001; Zeebe, 2014; Siegenthaler and Münnich, 1981; Clark and Lauriol, 1992). If this conversion step is furthermore mediated by NDH, this enzyme will likely discriminate against $^{13}C$ resulting in additional fractionation (Eichner et al., 2015). Overall, this 'internal C cycling' around NDH would yield higher $\varepsilon_p$ values than otherwise expected based on the $CO_2$ and $HCO_3^-$ fluxes over the
plasma membrane assuming equilibrium (Eichner et al., 2015; Sharkey and Berry, 1985).

Similar strategies may be present in the other algal groups that increase $\varepsilon_p$. Effective $CO_2$ fixation in diatoms relies on biophysical CCMs that facilitate or actively transport $CO_2$ and $HCO_3$ through a four-layered chloroplast membrane system (Keeling, 2013), which principally make the uptake more challenging but also confer additional control on the DIC fluxes (Matsuda et al., 2017; Nakajima et al., 2013). Numerous subcellular localized CAs are present in diatoms, which accelerate
the interconversion of $CO_2$ and $HCO_3^-$, also within the pyrenoid, where RubisCO is localized (Tachibana et al., 2011; Kikutani et al., 2016; Samukawa et al., 2014). Chemical disequilibrium environments between compartments, unidirectional conversion of CAs, or $^{13}C$ discrimination associated with $HCO_3^-$ transporters (solute carrier type transporters; SLC) may represent additional, but likely small sources of $^{13}C$ fractionation for diatoms.

No internal membrane systems with localized CAs associated to C fixation as present in diatoms have been recognized in
haptophytes and dinoflagellates (Rokitta et al., 2022). In fact, some dinoflagellate species even lack the pyrenoid compartment, where RubisCO is located in most eukaryotic algae (Ratti et al., 2007). The contribution of $HCO_3^-$ to photosynthesis is high in both groups (Rost et al., 2006; Rokitta and Rost, 2012; Bach et al., 2013; McClelland et al., 2017), and fractionation due to chemical disequilibria within the cell can therefore occur to some degree, e.g. by favoring unidirectional conversion of $CO_2$ to $HCO_3^-$ and vice versa. However, stronger internal C-cycling to maintain high $CO_2$ accumulation in proximity of RubisCO by
decreasing $CO_2$ leakage from the cell (Cassar et al., 2006; Schulz et al., 2007; Eichner et al., 2015; Hoins et al., 2016) and higher contribution of $HCO_3^-$ to net C fixation generally lead to higher build-up of $^{13}C$ within the cell (i.e., stronger internal Rayleigh fractionation) and consequently lower $\varepsilon_p$ values. Thus, while described modes of CCMs for the different groups are mostly in line with observed fractionation patterns (Schulz et al., 2007; Eichner et al., 2015; Hoins et al., 2016; McClelland et al., 2017), $\varepsilon_p$ values exceeding the intrinsic fractionation of RubisCO remain puzzling.





Wilkes and Pearson (2019) recently proposed that certain components of the CCM are differently regulated in nutrient-limited, light-replete cultures compared to light-controlled cultures, which could explain the often observed differences in $\varepsilon_p$ patterns between chemostat and dilute batch cultures (Fig. 4), and likewise reconcile why $\varepsilon_p$ values can exceed $\varepsilon_f$ estimates under some conditions. More specifically, the authors suggested that when cells are nutrient limited they can experience excess photon flux, which may be alleviated through fueling photocatalytic dehydration reactions of $HCO_3^-$ by internal CAs localized in the

thylakoid lumen. The acidic environment in the thylakoid favors the unidirectional conversion of $HCO_3^-$ to $CO_2$, while the alkaline environment in the chloroplast favors the unidirectional conversion of $CO_2$ to $HCO_3^-$. Light-induced stimulation of these processes may increase fractionation due to unidirectional hydration of $CO_2$ and dehydration of $HCO_3^-$ (up to ~25‰ and ~34‰, respectively; Wilkes and Pearson, 2019). However, the proposed $\varepsilon_p$ difference between light-limited and nutrient-limited cultures was not consistently found (Fig 4; Laws et al., 2001; Hoins et al., 2016a). Moreover, our results suggest this

"light-driven CCM" activity stems from the absence of a light-dark cycle during culture growth rather than from nutrient or light limitation (Fig. 1, 4b). This was especially the case for haptophytes and dinoflagellates, where $\varepsilon_p$ values were consistently elevated under continuous irradiance (Fig. 2). In addition to differences in light-dark cycle, other culturing variables also differed between the studies reviewed by Wilkes and Pearson (2019).

## 4.2 Experimental settings and resulting isotopic disequilibria

Studies yielding exceptionally high $\varepsilon_p$ values (apparently higher than $\varepsilon_f$) were, next to nutrient limitation and continuous irradiance, also performed in high biomass chemostats under continuous aeration with $CO_2$ (Fig. 4). In haptophytes, light-dark cycle could, next to POC production/$CO_2$, best explain differences in $\varepsilon_p$ (Table S2). The important role of light on fractionation was already discussed by Rost et al. (2002), and more recently highlighted by Phelps et al. (2021). They found that in coccolithophores, $\varepsilon_p$ depended more strongly on light intensity and light-dark cycle than on $CO_2$ concentrations (higher $\varepsilon_p$

values with more light exposure), even when corrected for C demands (Phelps et al., 2021; Rost et al., 2002). While continuous light led to higher $\varepsilon_p$ also in one dinoflagellate and several diatoms (Burkhardt et al., 1999b; Wilkes et al., 2017), the light-dependency of $\varepsilon_p$ seems strongest in haptophytes and may thus relate to changes in C flow between photosynthetic C fixation and calcification under changing light conditions (Krumhardt et al., 2017; Bolton and Stoll, 2013; Phelps et al., 2021). For instance, daylength had a significant influence on $\varepsilon_p$ of *E. huxleyi* (up to 8‰), as the preferred carbon source shifted from $CO_2$

under continuous light to $HCO_3^-$ uptake under light-dark cycles (Rost et al., 2002; Rost et al., 2006). However, not all phytoplankton species' fractionation responds similarly to changes in daylength. For two diatoms and one dinoflagellate species, for instance, $\varepsilon_p$ values were similar for cultures grown under continuous light or light-dark cycles (Burkhardt et al., 1999b).

In diatoms and dinoflagellates, the culturing approach or method of carbonate chemistry manipulation was, next to POC

production/$CO_2$, the best predictor for changes in $\varepsilon_p$ (Table S3, S4). Rost et al. (2008) pointed out that, next to aspects of the CCM itself, different carbonate chemistry manipulations and culturing methods can lead to different $CO_2$-dependencies between studies and different experimental setups.





Importantly, in the calculations for the $\delta^{13}C$ of $CO_2$ and thus also $\varepsilon_p$, chemical and isotopic equilibrium is assumed. In "open" carbonate chemistry systems with a continuous supply of $CO_2$, however, an equilibrium situation may not yet be reached before

phytoplankton assimilate carbon (Zeebe et al., 1999; Rost et al., 2008), which is even more so in case of high biomass cultures. Recent work from Zhang et al. (2022) showed that it takes much longer (several hours to days) for an isotopic equilibrium to be reached in empty algal culturing vessels than a chemical equilibrium and that this should be considered in $\varepsilon_p$ calculations. This discrepancy could lead to an overestimation of $\varepsilon_p$ in "open" carbonate chemistry systems compared to "closed" carbonate chemistry systems, which has been setup by pre-aeration with a certain $pCO_2$ or by acid/base additions, such as observed in

figure 4c. This may especially be true for cultures with high $CO_2$ treatments (high carbon supply) and high overall carbon demands (high biomass), as both favor disequilibria situations (see Fig. S3 for example of the dinoflagellate *Alexandrium tamarense*). In chemostats that were run with low cell densities, on the other hand, isotopic disequilibria may not play a role and therefore yield similar $\varepsilon_p$ values comparable to dilute batch studies (Fig. 4c). Hence, biases in $\varepsilon_p$ values introduced by isotopic disequilibria can be misinterpreted as treatment effects, e.g., as an effect of nitrate- vs light-limited growth. Moreover,

chemostat systems that are maintained with high biomass, even though they are meant to mimic oligotrophic systems, are not representative for natural environments as these system support only low biomass concentrations (Van de Waal et al., 2014).

## 4.3 Implications for proxies

The extent to which experiments reflect natural conditions is important regarding proxy development, as they feed the mechanistic model of $CO_2$-dependent carbon isotope fractionation and confounding factors. A more standardized approach in

performing these types of experiments, so representative natural light settings, using only one type of carbonate chemistry manipulation, and maintaining cultures at low biomass, would already substantially reduce variation in the $CO_2$-dependent $\varepsilon_p$ responses between studies (Fig. 3). Both species-specific differences and the effects of drivers (nutrient limitation, temperature conditions, etc.) would then be more straightforward to distinguish, as study design will not interfere as a concomitant source of $\varepsilon_p$ variation (Fig. 4). We note, however, that even when studies use comparable methods, findings can still vary. For example,

not all phytoplankton strains tested showed a negative relationship between $\varepsilon_p$ and POC production/$CO_2$ (Fig. 3), meaning that other environmental factors can mask the $CO_2$ dependence, which urges for caution when using $^{13}C$ fractionation during photosynthetic carbon fixation as a $CO_2$ proxy. Quantitative constraints on these confounding factors are crucial to guarantee that reconstructed signals exceed the related uncertainties.

Quantitative constraints on physiological variables, implied to be growth rate and cell geometry, but also membrane

permeability to $CO_2$, and the boundary layer thickness dependent on temperature, pH, and salinity, are in place in $CO_2$ proxy work with a catch-all term called *b* (Popp et al., 1998; Rau et al., 1996; Jasper and Hayes, 1990; Bolton et al., 2016; Stoll et al., 2019). The *b* value is often linearly correlated with modern dissolved reactive phosphate concentration in surface seawater (Bice et al., 2006; Bidigare et al., 1997; Pagani et al., 2005), as phosphate is a major nutrient that often co-limits with other important micronutrients such as iron, zinc, and cobalt, which affect phytoplankton growth rate and cell size (Bidigare et al.,

1997). However, this is quite a simplistic view and $b$ can vary substantially over time and between locations (Zhang et al. 2019).

Better constraints on $b$ may further advance $CO_2$ proxy development based on specific algal biomarkers, although this remains a challenge due to the sparsity of useful parameters on confounding factors in the paleo-environment. This is especially true for biomarker proxies, such as phytane and alkenones, as these biomarkers are produced by multiple species, certainly through

geological time (Witkowski et al., 2018). These species may have had different modes of carbon acquisition, growth rates and cellular carbon contents, and discrepancies between alkenone proxy and ice core records were also attributed to CCM activity (Badger et al., 2019; Badger, 2020). The preference for more specific proxy work is also needed as the analyzed phytoplankton groups show different slopes for the $\varepsilon_p$ versus POC production/$CO_2$ relationship (Fig. 2). However, a better selection of study sites (i.e., located in more productive ocean regions with possibly similarly responsive species as well) can reproduce $p$CO$_2$

estimates that are in agreement with the ice core records even with constant $b$ values (Zhang et al., 2019).

Another phytoplankton group and even species-specific line of $CO_2$ proxies under development is the [13]C fractionation of dinoflagellate resting cysts (Hoins et al., 2015; Sluijs et al., 2018). Single cysts can be analyzed with a recent analytical setup (van Roij et al., 2017), which provides the advantage that specific species can be selected so that an estimate of cell size can be made. Vegetative cell sizes of dinoflagellates generally correspond to sizes of their resting cysts (Finkel et al., 2007). Cyst

size may be used to infer cellular carbon contents, and together with phosphate concentrations for growth rate, can give a better estimate for carbon demands and therefore improve the constraints of $b$ in this line of proxies.

## 5 Conclusions

Our results illustrate that the POC production/$CO_2$-dependency of $\varepsilon_p$ can vary significantly between different phytoplankton species and groups, but also as a result of different culturing methods and differences in daylength, especially for haptophytes.

Extrapolating results to natural environments and for proxy applications therefore requires caution, and it should be carefully considered if culture methods and experimental conditions are representative of natural environments. Better approximations for carbon demands (described by $\mu_i$) in $\varepsilon_p$-based $CO_2$ proxies could also greatly improve their estimates. This will be challenging in the paleo-environment, especially with proxies that rely on biomarkers. Alternatively, careful selection of sites with more similar environments and phytoplankton species could also further improve proxy estimates.

## Data availability

All data presented in this study is available in the open data repository Dryad (DOI##)



**Author contributions**

All authors contributed to the study design. Preliminary data for the study was collected by MH. KB collected additional data, performed statistical analyses and wrote the first draft. All authors provided feedback on the manuscript.

**Competing interest**

The authors declare that they have no conflict of interest.

**Acknowledgement**

The authors thank Gert-Jan Reichart for constructive discussions that helped shape the presented ideas. AS thanks the European Research Council for Consolidator Grant 771497.

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





**Figures**

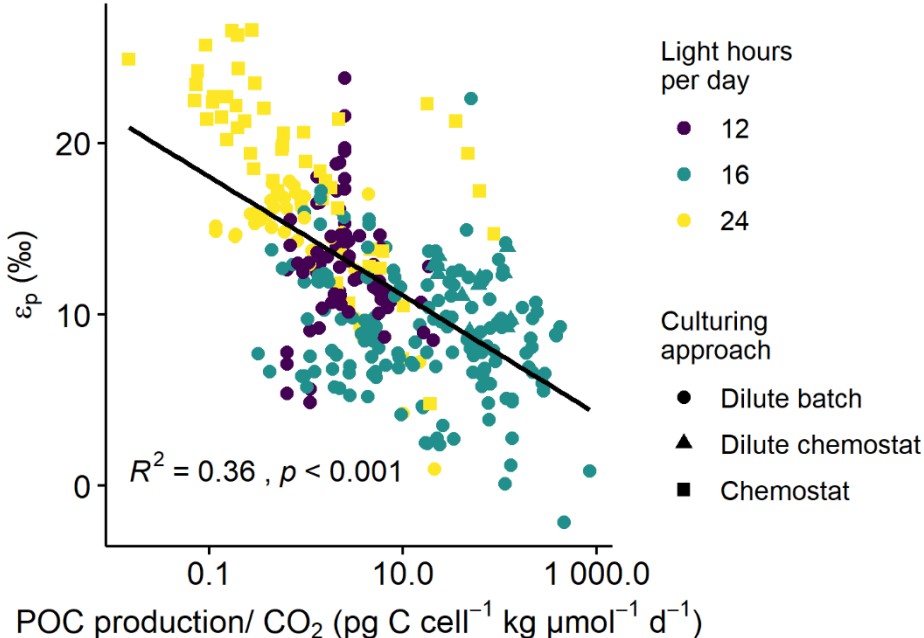

**Figure 1: POC production/CO₂ (C-demand/C-supply; log-transformed) against εₚ across all phytoplankton groups. Colors indicate the light-dark cycle; marker shapes indicate the culturing approach. Black line illustrates the log-linear relationship (R² and P indicated in the panel).**





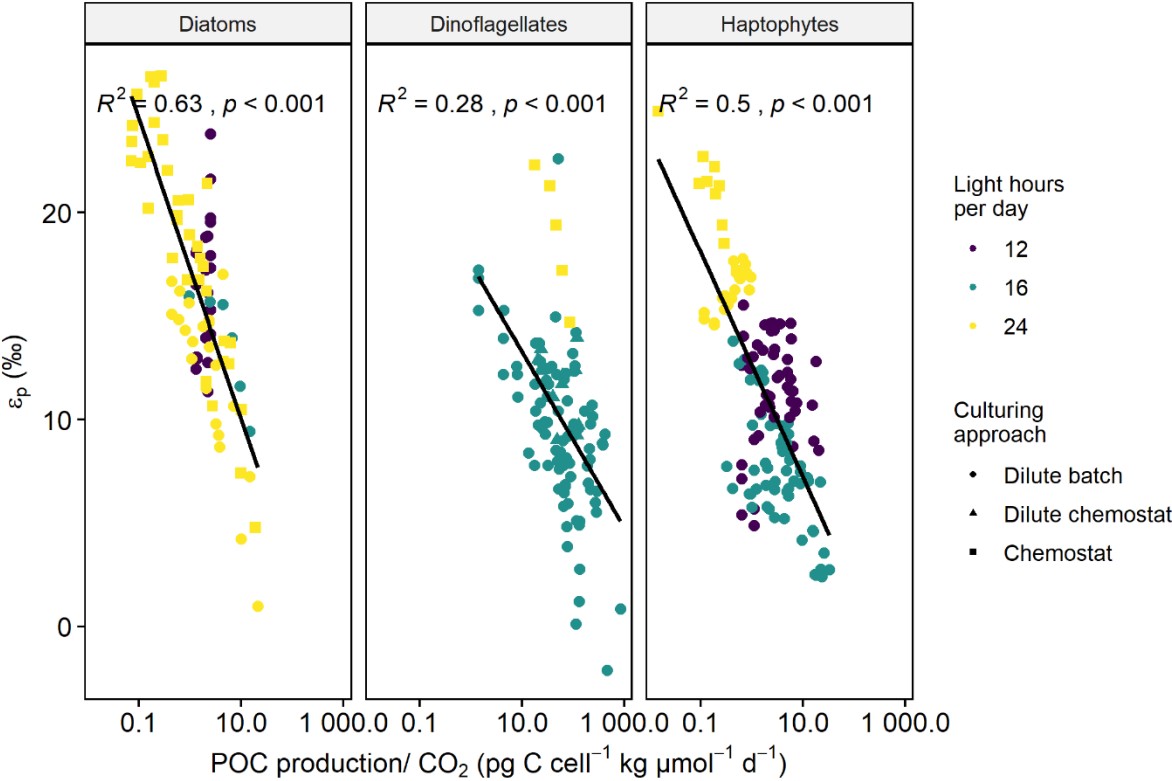


**Figure 2: POC production/CO₂ (carbon demand over supply; log-transformed) against ε$_p$ for the different phytoplankton groups, where the colored points indicate the respective light-dark cycle, and the shape of the points indicates the culturing approach. Black line illustrates the linear relationship (R$^2$ and P indicated in the panels).**

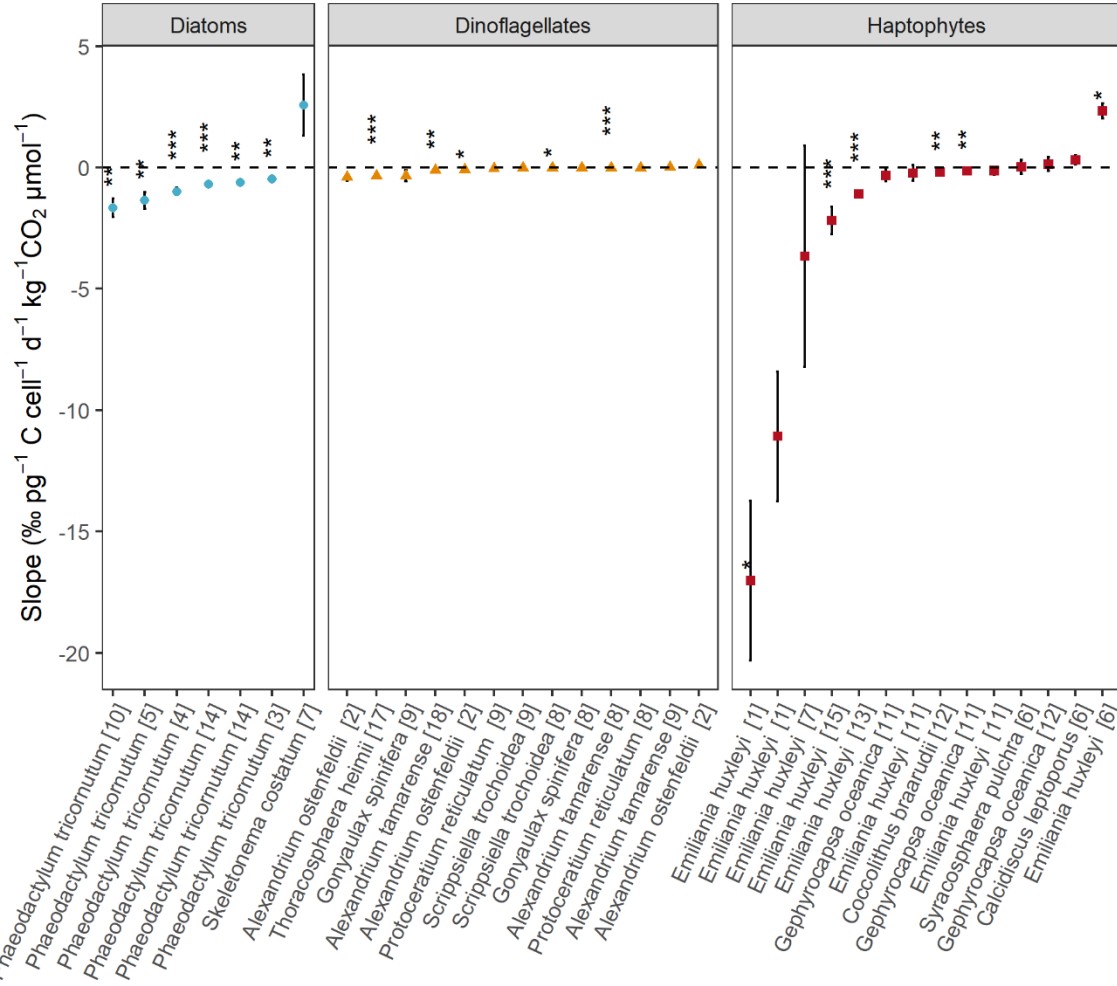


**Figure 3:** Slopes of $\varepsilon_p$ in response to POC production/$CO_2$ for the different species and studies using a linear fit. Numbers between brackets refer to the different studies (Table S1). Blue dots represent diatoms, orange triangles dinoflagellates, and red squares haptophytes. Significance is indicated by the asterisks (*** $P<0.001$, ** $P<0.01$, * $P<0.05$).





**Figure 4: Differences in ε$_p$ between a) type of carbonate chemistry manipulation (P-A is pre-aeration, and C-A is continuous aeration), b) light-dark cycle, where colors indicate the different phytoplankton groups and shapes indicate type of limitation (N is nitrogen, P is phosphorus, T is temperature, and No is non-limited), c) culturing approach, where colors indicate type of carbonate chemistry manipulations and shapes type of limitation again, and d) type of limitation, where colors indicate culturing approach and shapes light hours per day. Significant differences between experimental conditions are indicated by the letters.**

