# Peer review of "Physiological control on carbon isotope fractionation in marine phytoplankton"

_Biogeosciences, 2022_

## Author Response (AR1)

**Reply Reviewers**

**1**

Brandenburg et al. synthesize the available data on stable carbon isotope fractionation in phytoplankton to test how this parameter (Ep) is controlled by CO2 and other environmental parameters. This is a very well written and interesting paper and the data collection/analysis/interpretation seem very sound. I have hardly any comments, although I must admit that my knowledge on isotope fractionation is a bit rusted as I haven't followed the literature for a couple of years.

The key message of the paper is almost frustrating, nevertheless important. From my stand-point this paper requires only very minor revisions (but I hope the other Reviewer is more up to date on the topic than I am).

*We thank the reviewer for his/her kind words. In terms of proxy applications, the key message of our analysis is indeed discouraging. Below you will find our answers to the reviewer's comments and suggestions.*

Minor comments:

Figure 2: I found the unit of the C-demand/C-supply a bit strange. Wouldn't it be easier to keep the unit as for the individual components and put them in brackets i.e. (C-demand unit)/ (C-supply unit)? Just a suggestion to facilitate understanding what this parameter means.

*We agree with the reviewer that the units may be clearer, both in the figures and also in the text. Whenever we use POC production/CO2 concentration, we now consistently use POC production/[CO2]. We also made figure legends clearer, by typing the unit between brackets (e.g. (pg C cell$^{-1}$ d$^{-1}$) / (μmol kg$^{-1}$) like the reviewer suggested.*

Line 165: "…prevents diffusion of CO2 but is permeable for HCO3-…" This surprised me. Are is CO2 or HCO3 mixed up, perhaps? Just double-checking.

*No they are not mixed up. Cyanobacterial photosynthesis takes place in the carboxysome, and to allow sufficient build-up of $CO_2$ around RubisCO ensuring effective carboxylation, the membrane of this compartment is not permeable to $CO_2$. It is for $HCO_3^-$, which upon diffusion into the carboxysome is converted to $CO_2$. See also the references (i.e. Espie and Kimber, 2011).*

The supplementary material could be moved to the main text. I don't see a reason to bury it there.

*As we would like to keep to main text focused and condensed, we would prefer to keep the supplement as is.*

The figures are very well designed and informative.

*Thanks!*

**2**

The manuscript by Brandenburg et al. presents a compilation of stable carbon isotope fractionation data in phytoplankton experiments grown under various culture conditions (day length, nutrient availability, temperatures..) and teases out their contribution to the theoretically straightforward and expected demand/supply relationship on ep. The manuscript is an important contribution to the field, very well written and the data is presented nicely. I have therefore only a few comments and suggestions.

*We thank the reviewer for his/her kind words and will address the comments and suggestions below.*

General comments:

1) The authors should clarify the statistical approach. If I understood correctly, their linear models predicting ep had three factors, i.e. POC production/CO2, one influential condition (light, irradiance,.....), and species. While the influential condition and species factors have categorical or discrete factor levels, POC production/CO2 has not. Is that something the lmer function in R can handle? I was under the impression that all levels would need to be categorical or distinct (not a continuum without groups), as it is basically an ANOVA. Please clarify.

*The reviewer is correct that we used three predictor factors in our models (namely POC production/$CO_2$, one influential condition, and species). Lmer has no problem with fitting both continuous and discrete data as fixed predictor variables, as illustrated in the examples from Bates et al. (2015; https://cran.r-project.org/web/packages/lme4/vignettes/lmer.pdf). What the reviewer maybe refers to is that it does not make sense to use a continuous variable for generating a random intercept in R. However, we used only the discrete variable "species" to provide a random intercept, and no continuous variables, so our model structure is correct. Here is also the R syntax for clarification: lmer(Ep ~ POCproduction/CO2 +  [influential factor] + POCproduction/CO2: [influential factor]  + (1|Species), data). To make this more clear, we now specify that we used a random intercept also in the text (L112).*

2) The authors have chosen to test POC production/CO2 as the main driving factor for ep (please see also comment 1). From a pale-reconstruction perspective, that would require estimating two physiological parameters, i.e. POC per cell and instantaneous growth rate, to infer ep. What about the more simple growth rate/CO2 approach? The authors could test if they come to the same conclusions. I reckon they would but better to check.

*The reviewer is right that this more simple approach would be easier to apply from a paleo-reconstruction perspective. This is why we now include this analysis on instantaneous growth rate/CO2 in the supplementary ($\varepsilon_p$ versus $\mu_i$ /$[CO_2]$, Fig. S2 and S3) and also mention it in the main text (L126-130; 159-161). While these data confirm our conclusion, these figures clearly stress the need to also make estimations for cellular POC contents in the paleo-domain, as especially for haptophytes this makes a big difference with regard to the explanatory power.*

3) Again, from a reconstruction perspective, the authors could calculate how much explanatory power a multiple linear regression approach would generate. Of course, some of the factors would not work as being categorical (unless a generalised linear model would be used instead), but some

could be retained (e.g. light, temperature) or changed over (nutrient concentration, e.g. nitrate as being a proxy for the degree of limitation). That could be done group-specific, and looking at the simple linear regression presented in Figure 2, I could imagine that it would be quite a success.

*We thank the reviewer for his/her suggestion, and tested how much explanatory power we could generate with a multiple linear regression approach using different environmental variables (L114-116). As explained above, this approach can use both continuous and discrete data. We found that the inclusion of the light regime and whether there was nutrient limitation yielded highest explanatory power in all groups, and mention these findings now in the results (L131-133) and in the discussion (L156-157; 280-281).*

Specific comments:

1) L 241: either 'these systems' or 'this system'.

*Thanks for noting this mistake. We changed it accordingly.*